# Quantitative Correlation between Thermal Cycling and the Microstructures of X100 Pipeline Steel Laser-Welded Joints

**DOI:** 10.3390/ma13010121

**Published:** 2019-12-26

**Authors:** Gang Wang, Jinzhao Wang, Limeng Yin, Huiqin Hu, Zongxiang Yao

**Affiliations:** 1School of Metallurgy and Materials Engineering, Chongqing University of Science and Technology, Chongqing 401331, China; wwg_16@163.com (G.W.); yaozongx@163.com (Z.Y.); 2Guangdong Provincial Key Laboratory of Advanced Welding Technology, Guangdong Welding Institute (China-Ukraine E.O. Paton Institute of Welding), Guangzhou 510650, China; jinzhao_wang@foxmail.com; 3School of Natural and Applied Sciences, Northwestern Polytechnical University, Xian 710129, China; huiqin_hu@foxmail.com

**Keywords:** laser welding, numerical simulation, X100 pipeline steel, welding thermal cycle, microstructure

## Abstract

Due to the limitations of the energy density and penetration ability of arc welding technology for long-distance pipelines, the deterioration of the microstructures in the coarse-grained heat-affected zone (HAZ) in welded joints in large-diameter, thick-walled pipeline steel leads to insufficient strength and toughness in these joints, which strongly affect the service reliability and durability of oil and gas pipelines. Therefore, high-energy-beam welding is introduced for pipeline steel welding to reduce pipeline construction costs and improve the efficiency and safety of oil and gas transportation. In the present work, two pieces of X100 pipeline steel plates with thicknesses of 12.8 mm were welded by a high-power robot laser-welding platform. The quantitative correlation between thermal cycling and the microstructure of the welded joint was studied using numerical simulation of the welding temperature field, optical microscopy (OM), and scanning electron microscopy (SEM) with energy-dispersive spectroscopy (EDS). The results show that the heat-source model of a Gaussian-distributed rotating body and the austenitization degree parameters are highly accurate in simulating the welding temperature field and characterizing the austenitization degree. The effects of austenitization are more significant than those of the cooling rate on the final microstructures of the laser-welded joint. The microstructure of the X100 pipeline steel in the HAZ is mainly composed of acicular ferrite (AF), granular bainite (GB), and bainitic ferrite (BF). However, small amounts of lath martensite (LM), upper bainite (UB), and the bulk microstructure are found in the columnar zone of the weld. The aim of this paper is to provide scientific guidance and a reference for the simulation of the temperature field during high-energy-beam laser welding and to study and formulate the laser-welding process for X100 pipeline steel.

## 1. Introduction

As a result of the immense consumption of oil and gas energy driven by the rapid development of the world economy, the scope of energy exploration is constantly expanding, and oil gas fields are usually far from the end of the consumer market. The use of high-grade steel pipelines with large pipe diameters, thick walls, and high strength and toughness substantially improves the cost-effectiveness of construction to ensure transport efficiency and safety [1,2,3].

The efficiency and quality requirements for welding in the installation of large-diameter, thick-walled pipes in harsh environments and over long distances will become more stringent. Therefore, many advanced welding methods are used for welding high-grade pipeline steel in recent years, such as laser welding [4], laser-arc hybrid welding [5,6], friction stir welding [7], and more. Laser welding has good application prospects due to its advantages such as fast welding speed, large penetration depth, and small deformation, and has been reported in many welding fields. Guo et al. [8] reported the microstructure and mechanical properties of underwater laser welding of Titanium alloy. Xu et al. [9] as well as Xin et al. [10] investigated the characteristics and process mechanism of laser welding or laser-arc hybrid welding of aluminum alloys. Zhou et al. [11], Chen et al. [12], and Casalino et al. [13] studied the laser welding-brazing process of Titanium-Aluminum dissimilar metals. Silva Leite et al. [14] and Hipp et al. [15] discussed the laser welding process and mechanism of various stainless steels.

However, the final microstructure of high-grade pipeline steel in a laser-welded joint depends on the temperature field formed by the laser heat source and on the influence of thermal cycling during the welding process, which directly determines the strength, toughness, and service reliability of a welded joint [16,17,18,19]. Therefore, to accurately control the strength and toughness of a laser-welded joint in a high-grade long-distance pipeline and to improve the quality, efficiency, and cost benefits of pipeline installation, building the quantitative correlation between the thermal cycling and the microstructure of high-grade pipeline steel within laser-welded joints is important [2,16,20]. Different thermal cycling leads to different microstructures within welded joints and, accordingly, to different strengths and toughness in these welded joints [17,21,22,23].

In this research, based on the actual working conditions of an X100 pipeline steel plate with a thickness of 12.8 mm, reasonable and feasible process parameters are chosen to obtain high-quality welded joints. The microstructures of the metallographic specimens obtained from the laser-welded joints were analyzed by optical microscopy (OM) and scanning electron microscopy (SEM) in conjunction with energy-dispersive X-ray spectrometry (EDS). In addition, a finite-element temperature field simulation was carried out using the parameters of the thermal physical properties of X100 pipeline steel, an accurate self-developed laser-welding heat-source model, a reasonably simplified finite-element mesh, and welding temperature field initialization conditions. Ultimately, to reveal the quantitative correlation between thermal cycling and the microstructures of X100 pipeline steel laser-welded joints, the intrinsic correlation between the temperature field of the laser welding and the microstructure within the welded joint was analyzed and discussed on the basis of metallurgical and phase-change theories.

## 2. Experimental Procedures for X100 Laser Welding

### 2.1. Experimental Materials

The composition of the X100 pipeline steel plate with a thickness of 12.8 mm that conformed to the API-5L standard is shown in Table 1. The production of X100 pipeline steel is based on alloying technology and the thermomechanical control process (TMCP), which compensates for the loss of strength caused by the reduction of the carbon content from adding alloying elements and improves the comprehensive properties of the steel via alloy phase-transformation strengthening, precipitation strengthening, and fine-grain strengthening [24,25,26]. The original microstructure of the X100 pipeline steel is mainly bainite, including granular bainite (GB), acicular ferrite (AF), and martensite-austenite (M-A) constituent. A large amount of the M-A constituent is distributed between or within grains, and the dislocation density of the lattice distortions increases due to the martensitic transformation. Transformed martensite is also the location of microcracks [27,28]. Therefore, controlling the shape and distribution of the M-A constituent in laser-welded joints of X100 pipeline steel is important [25,27,29,30].

### 2.2. Experimental Methods

The processing equipment consisted of a laser-welding head with TRUMPF BEO D70 90° focusing optics (200-mm collimation focal length, 200-mm focus length, and 200-μm transmission fiber) mounted on a precision six-axis robotic arm (KUKA KR 60 HA). The robotic arm was configured to perform the necessary motions required for welding as per the welding procedure specification (WPS) through laser-welding control software. The high-energy laser beam was supplied by a disc laser (TRUMPF TruDisk 10002, TRUMPF, Ditzingen, Germany) with a maximum output power of 10 kW and was protected by a gas protection device (Figure 1) during welding.

The welding specimens were thoroughly cleaned prior to welding. The welding was performed in a flat position (PA) on both sides. The process parameters employed for the laser welding are given in Table 2. The laser power is identified based on the penetration capacity of the laser welding. The other parameters are determined by the orthogonal experiments. The welding of the X100 steel specimens was performed in the I-groove joint configuration. The metallographic samples (30 mm × 20 mm × 12.8 mm) for the microstructural examinations were machined by wire electric discharge machining (WEDM). The machined microstructural samples were then polished following the standard microstructural examination procedure and were etched using a 4% Nital solution (for 3–8 s). The bead shape measurements and the microstructural examinations were performed using OM (Axio Imager M2m, ZEISS, Oberkochen, Germany) and SEM (Nova NanoSEM 430, FEI, Hillsboro, OR, USA). OM was performed on a metallurgical microscope. SEM was performed on a scanning electron microscope (Hitachi S-3700N). The quantitative analyses of the micrographic images were performed using image analysis software (Image-Pro Plus).

## 3. Numerical Calculation of the Laser-Welding Temperature Field

### 3.1. Establishment and Solution of the Finite-Element Model

The Multiphysics Object-Oriented Simulation Environment (MOOSE), which is an open-source finite-element analysis software framework developed and maintained by Idaho National Laboratory, was used for the X100 pipeline steel laser-welding temperature field simulation. To compare the laser-welding temperature field simulation results with the actual working conditions of the welding process, a two-dimensional finite-element model consistent with the actual welding joint size was established through reasonable model abstraction and reliable data for the material thermophysical parameters that were calculated by JmatPro [31,32,33,34,35]. The material thermophysical parameters are shown in Table 3.

The model size was 10 mm × 12.8 mm, and the mesh size was 0.05 mm × 0.05 mm. According to the shape parameters of the weld seam and the welding heat-affected zone (HAZ), the heat-source model for a Gaussian-distributed rotating body is proposed (Equations (1)–(3)). Heat convection, heat conduction, and heat radiation are taken into account in the boundary conditions (Equations (4) and (5)) [32,34,36,37,38,39,40]. The model was solved by the Preconditioned Jacobian-Free Newton-Krylov algorithm (PJFNK). The linear tolerance was 1 × 10^−2^, the nonlinear absolute tolerance was 1 × 10^−8^, and the time step was 0.01 s.

Heat-source model
(1)$$Q\left(x,y,z\right)=\frac{\eta_{L}P}{V_{hs}}exp\left(-\frac{3\left({\left(x-vt\right)}^{2}+y^{2}\right)}{R^{2}}\right)$$
(2)$$R=R_{0}\left(0.2sin\left(\frac{4\pi z}{H}\right)+\frac{z^{\frac{2}{5}}}{H}\right)$$
(3)$$V_{hs}=\pi \int_{0}^{H}R^{2}dz$$

Governing equation
(4)$$\left\{k\left(T\right)\left(\frac{\partial^{2}T}{\partial x^{2}}+\frac{\partial^{2}T}{\partial y^{2}}+\frac{\partial^{2}T}{\partial z^{2}}\right)\right\}+Q=\rho \left(T\right)C_{p}\left(T\right)\left(\frac{\partial T}{\partial t}\right)$$

Boundary condition
(5)$$k\frac{\partial T}{\partial r}=-h_{A}\left(T-T_{ref}\right)-\sigma_{s}\varepsilon_{r}\left(T^{4}-T_{ref}^{4}\right)-q_{con}$$
where *Q*(*x*, *y*, *z*) is the heat flux density at point (*x*, *y*, *z*), $\eta_{L}$ is the thermal efficiency of laser welding, P is the laser power (W), ***v*** is welding speed (mm/s), *t* is welding time (s), $R_{0}$ is the radius of the laser welding keyhole (mm), H is the height of the heat source (mm), k is the thermal conductivity of X100 pipeline steel, ρ is the density, $C_{p}$ is the specific heat, *T* is the temperature (K), $h_{A}$ is the convective heat transfer coefficient on the test plate surface, $T_{ref}$ is the ambient temperature (K), $\sigma_{s}$ is the Stefan-Boltzmann constant, $\varepsilon_{r}$ is the emissivity of the blackbody, and $q_{con}$ is the energy density of heat conduction. In order to improve the numerical calculation efficiency of the welding temperature field, the three heat transfer effects are simplified as a parameter of the comprehensive convection heat transfer coefficient. For the numerical calculation of laser welding temperature of X100 pipeline steel $\eta_{L}$ = 0.85, *v* = 35 mm/s, *P* = 10,000 W, H = 10 mm, $R_{0}$ = 1.12 mm, and $T_{ref}$ = 300 K. In addition, the comprehensive convection heat transfer coefficient was 0.2 W/(m^2^·K) [37,38,39,41,42].

### 3.2. Post-Processing of Welding Temperature Field Data

The results calculated for the laser-welding temperature field were output in the EXODUS II format. The welding-temperature field contour, the welding thermal cycle curve, the *t_85_* field contour, and the weld-pool section profile were obtained from the temperature field data through data visualization software (Paraview, Kitware, New York, NY, USA) and MATLAB (MathWorks, Natick, MA, USA)). The simulated results for the laser-welding temperature field are in good agreement with the experimental results, as shown in Figure 2b.

According to the symmetry of the microstructural state of the laser-welded joint, the position 2 mm from the upper surface of the welding plate on the cross section of the weld is taken as the focus of attention. As shown in the diagram in Figure 2a, we extract the welding heat cycle curve in the grid node from the base material of the weld and calculate the *t_8/5_* value, which is used for analysis of the microstructural distribution in the laser-welded head.

## 4. Results and Discussion

### 4.1. Extraction of the Characteristic Parameters of the Welding Thermal Cycle

Because the thermal cycling process of a welded joint determines its microstructure, deep analysis of the thermal cycling process of the X100 pipeline steel is beneficial for the determination and analysis of the microstructure. The laser-welded joint of X100 pipeline steel consists of the base material, a partial-phase-transition zone, a normalized zone, an overheated zone, and a melted zone from the base material to the weld, according to the process of thermal cycling. The corresponding microstructures are the base material, a banded-microstructural zone, a fine-grain zone, a coarse grain zone, and a columnar zone.

The maximum temperature field (Figure 3a) and the *t*_8/5_ field (Figure 3b) are obtained by post-processing the numerical results of the laser-welding temperature field. The *t*_8/5_ field, just like the temperature field, is a general term for the distribution of *t*_8/5_ at all points in the welded joint. The welding HAZ exhibits a clear temperature gradient, even though the change in *t*_8/5_ is relatively small. However, the change in the maximum temperatures of the weld is relatively small, and the gradient of change in *t*_8/5_ is clear.

The maximum temperature ranges are observed in each microstructure zone, and the variation in the welding thermal cycle for each zone is relatively small. Therefore, the average value curve is taken as the characteristic thermal cycle curve of the zone. As shown in Figure 4, the characteristic thermal cycle curve is surrounded by a heated region with room temperature *T*_0_ and peak temperature *T_max_*. The heating time is *T_max_*–*T*_0_. The high-temperature zone is surrounded by the *T*_800_ curve, and the high-temperature residence time is *T*_800_*^max^*–*T*_800_*^min^*. The cooling zone is surrounded by *t*_800_ and *T*_500_, and the cooling time is *t*_500_-*t*_800_.

The original microstructure of the X100 pipeline steel underwent austenitization and continuous cooling of the undercooled austenite during the welding thermal cycling. The degree of austenitization is determined by the heating rate, the peak temperature, and the residence time at a high temperature. The parameterization can be defined as what corresponds to the region enclosed by the characteristic thermal cycling curve *T*_0_ and *t*_800_ in Figure 5.
(6)$$D_{A}=\int_{t_{0}}^{t_{800}}T\left(x,y,z,t\right)-T_{0}dt$$

The function T(x,y,z,t) of temperature (*T*) with time (*t*) at point (*x*, *y*, *z*) in a welded joint is called the welding thermal cycle curve. Since *t*_8/5_ determines the cooling transformation process of the undercooled austenite, defined as *t*_8/5_
*= t*_500_
*− t*_800_, the average cooling rate of the corresponding position is *v_c_* = (800 − 500)/*t*_8/5_.

The driving force for austenitization in the X100 pipeline steel is the difference between the volume-free energies of the original microstructure and the austenite. The austenitization process can be divided into four stages: (1) nucleation, (2) nucleus growth, (3) dissolution of residual carbides, and (4) homogenization of austenite, which is affected by the heating rate, the peak temperature, and the residence time at high temperature. Generally, austenite nucleates inhomogeneously on the subcrystal boundaries of bainite and ferrite grains or on carbonitrides of the alloying elements, which forms two new interfaces after nucleation. It grows through the transformation of ferrite to austenite and the dissolution of the carbonitrides [24,25,26,27,28,43]. The rate of formation of austenite depends on the nucleation rate and the growth rate. The rate of formation of the austenite and the inhomogeneity of the austenite composition increases with a growing heating rate. The initial austenite grains are refined with the heating rate [22,25,30,44,45].

Alloying elements affect the formation rate of austenite, the dissolution of carbides, and the homogenization of austenite. The diffusion coefficient of carbon is affected by alloying elements. In addition, the diffusion coefficient of carbon in austenite is reduced, and the formation rate of austenite is slowed by strong carbides, such as those of Cr, Mo, W, V, Ti, and Nb [43,44,45,46,47].

The cooling rate determines the continuous cooling transformation of undercooled austenite, which determines the final structure and mechanical properties of a laser-welded joint in X100 pipeline steel. With increasing cooling rates, the diffusion of carbon and alloying elements decreases gradually. The microstructure transformation types are diffusive phase transition, semi-diffusive phase transition, block transformation, and shear transformation. The products of the transformations are ferrite, GB, upper bainite, M-A components, Weiss structures, BF, and lath martensite.

### 4.2. Microstructure and Quantification of Welded Joints

The laser-welded joint of X100 pipeline steel exhibits a clear microstructure gradient from the base metal to the weld, which can be divided into the following sections, as shown in Figure 5. The sections include the base metal (BM), a banded-microstructure heat-affected zone (BMHAZ), a fine-grained heat-affected zone (FGHAZ), a transitional-microstructure heat-affected zone (TMHAZ), a coarse-grained heat-affected zone (CGHAZ), and a columnar zone.

The microstructure of the Xl00 pipeline steel in the laser-welded joint mainly includes the AF, GB, BF, and M-A constituents. The relative proportions of these four microstructural components have important effects on the properties of the joint. The M-A constituent from carbon-enriched and untransformed austenite surrounded by bainite is known to deteriorate the toughness of bainitic steel [24,25,27,30]. In addition, when the AF and GB contents in Xl00 pipeline steel are too low, the original austenite grains are not separated, and the growth of the lath beams is not controlled, which results in large effective grain sizes for the lath bundles and diminished elongation and impact properties. When the AF and GB contents are too high, the strength of the steel plate is greatly reduced. AF and GB easily nucleate and grow along grain boundaries, which segments the original austenite grains so that clear substructures appear in the deformed grains near the grain boundaries in addition to the original large-angle boundaries. The original austenite grains are divided into numerous regions with different orientations and uneven sizes by the AF and GB. The orientations in the region are approximately the same, and the orientation differences are small. The growth of the lath beams is limited in the segmented region, and the AF and GB microstructures refine the grains through the phase-transformation processes. The GB and LB contents decrease with increasing cooling rate, whereas the AF content shows the opposite effect [24,27,28,47,48]. Furthermore, the decrease in the amount of brittle and hard phases in steel leads to a decrease in the strength.

The composition and microstructure segregations in X100 under thermomechanically controlled processing are the fundamental reasons for the formation of banded microstructures in laser-welded joints. According to a comparative analysis of the microstructures observed in the SEM micrographs and those observed in the OM images, the banded microstructure is composed of bainite bands and M-A bands. The GB in the bainite bands is smaller than in the base material, and the solid-solution alloying elements are partially precipitated, which indicates that fewer grain defects are present, the overall energy is lower, and the corresponding BM area is more stable. The M-A bands form in the microstructure of the BM, which is characterized by a small amount of alloying elements, the strong aggregation of the alloying elements, and a relatively high energy stability. The bainite band appears as a dark band under an optical microscope, and the granular M-A constituent is clearly observed via high-powered SEM. With a decreasing distance from the center of the weld, the bainite content decreases gradually (Figure 6b).

OM revealed a large gradient from fine grains to coarse grains in the HAZ. With a decreasing distance from the center of the weld, the grain size of the original austenite gradually increases and the white bainite microstructure and the grain size gradually increase. Combined with the SEM analysis results, these results reveal that the grain boundaries of primary austenite are clear and that a small amount of M-A constituent exists at the grain boundary, which consists mainly of a small amount of GB dispersed within BF (Figure 6c–e). However, the coarse-grained area near the fusion line is relatively coarse. The sizes of the original austenite grains and GB increase. The block bainite appears and the BF lath length increases. In addition, a small amount of lath martensite is present.

The weld of the laser-welded joint of X100 pipeline steel mainly consists of coarse columnar crystals. In addition, large bulk bainitic microstructures are present, the BF lath beams are longer, and a large-angle relationship exists between the laths (Figure 6f).

According to the EDS results, the carbon contents of the carbon-rich bainite and lath martensite in the BF and the lath martensite are approximately 1.18% and 0.72%, respectively. As shown in Figure 7d, the solid-solution alloying elements in the BF and martensite are Cr, Mn, and Si.

According to the comparison between the microstructure analyses and the energy-dispersive spectrometer analyses of the X100 pipeline steel, the microstructure in the HAZ is mainly granular bainite and acicular ferrite. As shown in Figure 6b–e, the dark white matrix is acicular ferrite, and the bright white granular microstructure is granular bainite. The microstructure of the weld is relatively complex. For example, the bright white granular microstructure in Figure 7a is the M-A constituent, and the bainite matrix in Figure 7b is bainitic ferrite.

### 4.3. Quantitative Relationship between Thermal Cycling and the Microstructure

The microstructures of welded joints are determined by the microstructure transformations from the welding thermal cycle. The degree of austenitization determines the starting state for the continuous cooling transformation of undercooled austenite, and *t*_8/5_ determines the type of continuous cooling transformation for the undercooled austenite, which leads to differences in the products of the microstructure transformations.

As shown in Figure 8, when the distance from the center of the weld increases, the degree of austenitization (austenitization index *D_A_*: 433.4–92.8) decreases linearly, and *t*_8/5_ (1.3–0.6 s) increases exponentially. The corresponding cooling rate decreases sharply. From BM to WELD, the microstructure changes from AF, GB, and MA to GB, MA, and BF, and the transformation occurs in the TMHAZ. From BM to FGHAZ, the AF:GB:MA ratio changes from 9.16:69.03:6.06 in BM to 2.04:75.4:18.58 in BMHAZ and to 2.22:60.35:17.5 in FGHAZ. From TMHAZ to WELD, GB:MA:BF changes from 4.8:11.84:83.36 in TMHAZ to 11.74:7.9:80.36 in CGHAZ and to 2.17:6.7:82.47 in WELD.

From the symmetry of the temperature field contours and the *t*_8/5_ contours, the austenitization degree index and the *t*_8/5_ distribution of the X100 laser-welded joint can be deduced to be equivalent. Thus, the microstructure of the welded joint from BM to WELD is also consistent with the characteristic region studied.

## 5. Conclusions

(1)Accurate heat-source model parameters are the core of the accurate simulation of temperature fields resulting from the laser-welding process. The Gaussian-distributed rotating body heat-source model presented in this paper accurately simulates the welding temperature field, and the calculation process is stable and reliable.(2)The final microstructure, which determines the strength and toughness of a laser-welded joint, is determined by the welding thermal cycle process, which affects the austenitization and continuous cooling transformation of the original microstructure of the BM. The degree of austenitization is more significant than the cooling rate.(3)The degree of austenitization in the microstructures at different positions in welded joints can be expressed by the parameter $D_{A}=\int_{t_{0}}^{t_{800}}T\left(x,y,z,t\right)-T_{0}dt$, which accurately reflects the effects of the heating rate, the peak temperature, and the residence time at high temperature on the original microstructure of the BM in the heated region and in the high-temperature region.(4)With a decreasing distance from the center of the weld and the upper surface of the weld, the heating rate, the peak temperature, and the residence time at high temperature of the welding heat cycle increase gradually. The austenitization degree and the cooling rate increase significantly, which results in increasing grain sizes for the original austenite. However, the final microstructure has less austenite overall.(5)The microstructure of the X100 pipeline steel in the HAZ is mainly composed of AF, GB, and BF, with small amounts of lath martensite, upper bainite, and bulk microstructure found in the columnar zone of the weld.(6)The microstructure gradient of the HAZ of the welded joint of X100 high-grade pipeline steel is mainly affected by the different degrees of austenitization in the BM and in locations where the change in the cooling rate is relatively small. The contribution of the cooling rate to the microstructure transition from FGHAZ to CGHAZ is limited.(7)Welding the same material under the same austenitization degree and cooling rate yields the same final microstructure and phase proportions, and the strength and toughness properties of the welded joints are the same.

## Figures and Tables

**Figure 1 materials-13-00121-f001:**
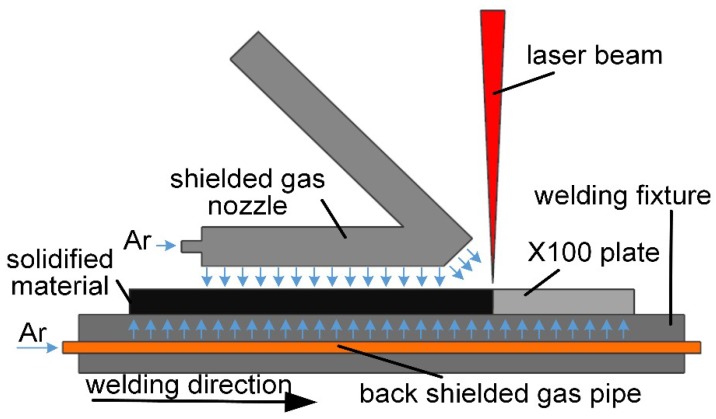
Schematic of the X100 laser-welding device.

**Figure 2 materials-13-00121-f002:**
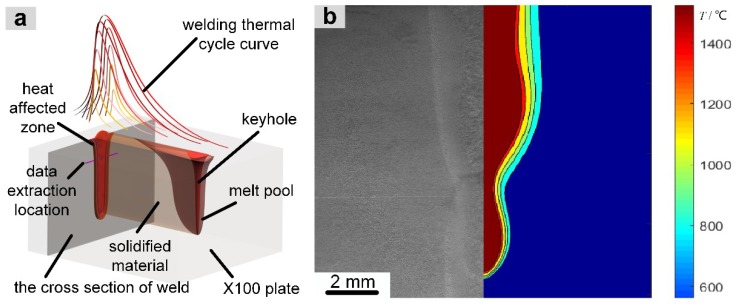
Principle of laser welding (**a**) and comparison of the temperature field simulation and experimental results (**b**) for X100 pipeline steel.

**Figure 3 materials-13-00121-f003:**
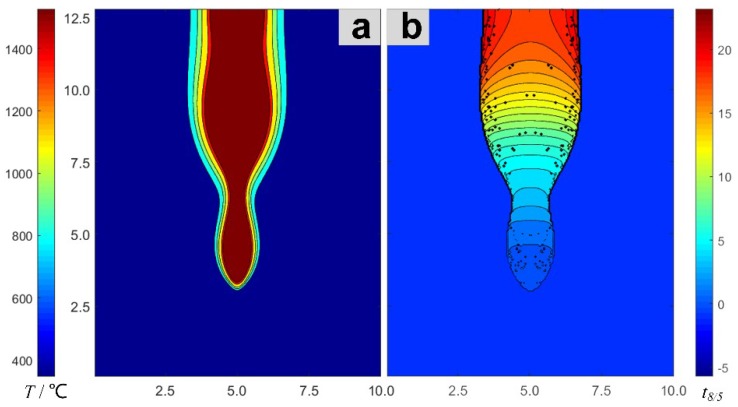
Maximum temperature field (**a**) and *t*_8/5_ field (**b**) of the weld section.

**Figure 4 materials-13-00121-f004:**
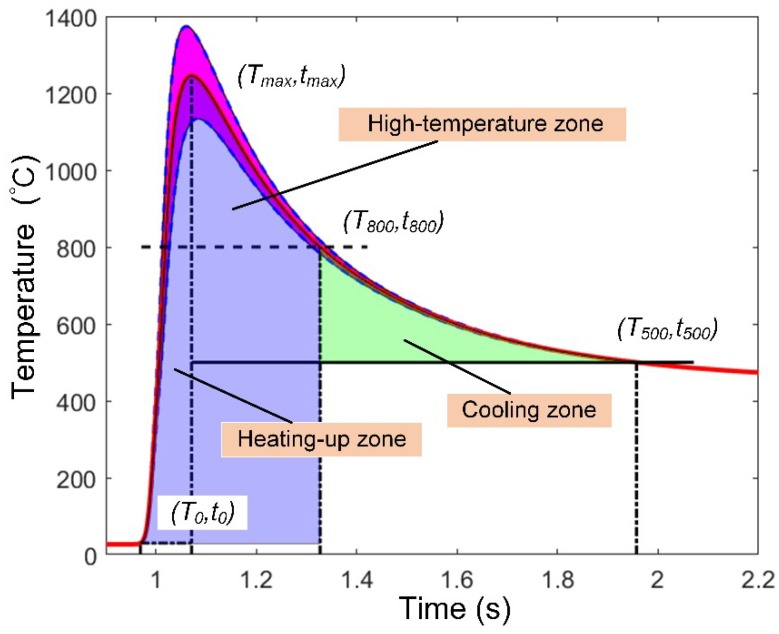
Thermal cycle curve and mean value curve of the laser-welding temperature field in the coarse-grained heat-affected zone.

**Figure 5 materials-13-00121-f005:**
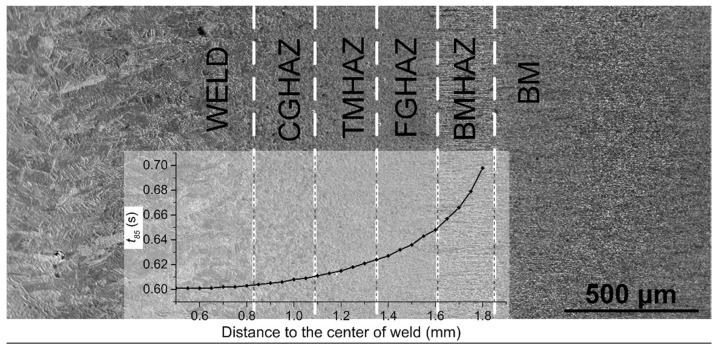
Microstructural zones and the trend of *t*_8/5_ in the X100 laser-welded joint.

**Figure 6 materials-13-00121-f006:**
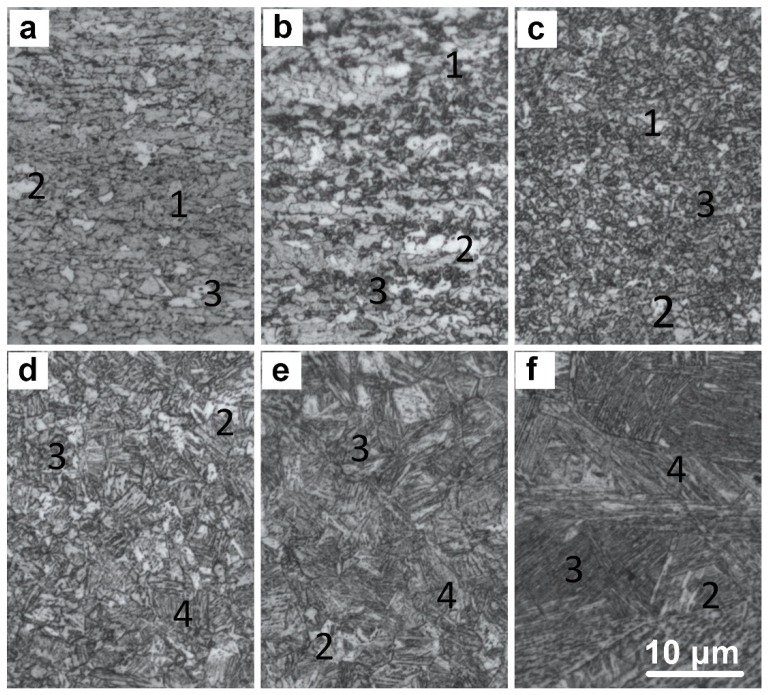
Optical images of the microstructures in the X100 laser-welded joint (1-AF, 2-GB, 3-MA, 4-BF): (**a**) BM, (**b**) BMHAZ, (**c**) FGHAZ, (**d**) TMHAZ, (**e**) CGHAZ, and (**f**) WELD.

**Figure 7 materials-13-00121-f007:**
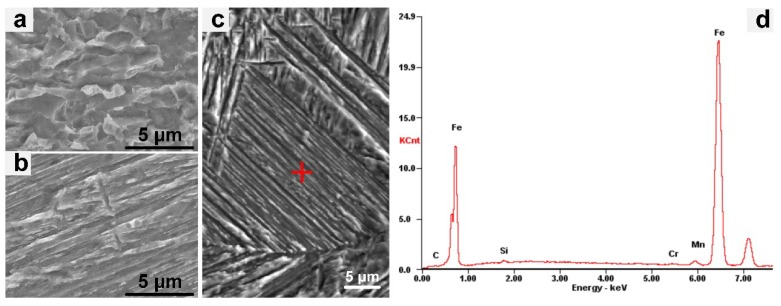
(**a**–**c**) X100 characteristic microstructures (SEM) of the laser-welded joint and (**d**) the energy spectrum of bainitic ferrite (BF).

**Figure 8 materials-13-00121-f008:**
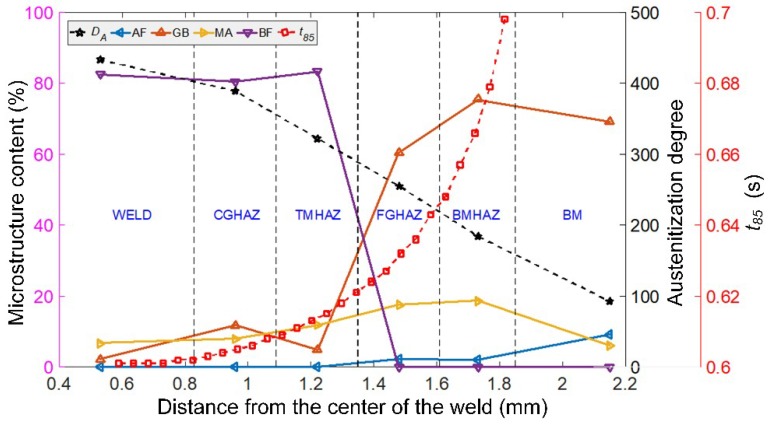
Quantitative relationship between the degree of austenitization and *t*_8/5_ and the microstructure distribution in the laser-welded joint of X100 pipeline steel.

**Table 1 materials-13-00121-t001:** Chemical composition of X100 pipeline steel (wt.%).

AL	C	CO	CR	CU	FE	MN	MO	N	NB	NI	P	SI	TI	V
0.012	0.064	0.003	0.023	0.28	96.90	1.87	0.003	0.017	0.017	0.47	0.009	0.099	0.017	0.002

**Table 2 materials-13-00121-t002:** Parameters for the laser welding of the pipeline steel.

Material (mm)	Laser Power (kw)	Welding Speed (m/s)	Amount of Defocusing (mm)	Frontal Protective Gas Flow (L/min)	Back Protective Gas Flow (L/min)
X100 (12.8)	10	0.035	−4	15	25
	8	0.035	−4	15	25

**Table 3 materials-13-00121-t003:** Thermophysical parameters of X100 high-grade pipeline steel.

Temperature (K)	293	373	473	673	873	1073	1273	1473	1773
Density (kg/m^3^)	7810	7790	7770	7720	7650	7610	7560	7500	7500
Thermal Conductivity [W/(m·K)]	54.42	54.01	52.75	52.75	34.67	27.55	23.18	21.54	21.54
Specific Heat [J/(kg·K)]	423	493	536	662	827	614	565	516	516

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
