# Peer review of "Quantitative Correlation between Thermal Cycling and the Microstructures of X100 Pipeline Steel Laser-Welded Joints"

_materials, 2019, doi:10.3390/ma13010121_

Round 1
Reviewer 1 Report
The article is devoted to the urgent problem of quantifying the correlation between thermal cycling and the microstructure of welded joints with X100 steel pipelines. The article is of interest primarily from the standpoint of numerical modeling, as well as experimental work.
Specific comments:
1. The authors need to give explanations to equations 1-5, write down the variables and indicate their values adopted during the simulation.
2. The final part of the introduction should contain the objectives of the study, and the conclusions reached as a result of their solution are presented in the conclusion section.
3. In figure 8, it is advisable to give an explanation of each curve.
Reviewer 2 Report
An article presents original investigations of the microstructure of laser welded joints. In general, the manuscript is well written. However, there are some minor issues that must be explained or corrected. I am convinced that with proper corrections the manuscript can satisfy the publication criteria in Materials.
Detailed comments:
According to the abstract, the Authors used X100 pipeline steel. However, Table 1 presents the chemical composition of X80 pipeline steel. Why?
Line 78: Rewrite "17-18" as "17,18".
One material (X100 steel" is used in investigations. So, in the title of section 2.1 the plural form is used?
I think that it will be better to combine the section 2.2 and 2.3 in one section.
line 85: When referencing tables and figures use "Figure", "Table" instead of abbreviations "Fig.", "Tab.". Please check the entire manuscript and see also Instructions for Authors.
line 90: The genesis of abbreviation (PA) is not clear for me. (PA) corresponds to the "position" or "flat position"?
According to the Table 2, only two sets of welding parameters are investigated. The difference between them is only laser power. The statement in line 92 that "These parameters were identified through a set of rigorous experiments." is not sufficient. The detailed procedure to find optimal welding parameters must be added. Methods should be described with sufficient details to allow others to replicate and build on published results.
line 103: Check grammar in the title of section 3.1.
The equations 1-5 are well known in the numerical analysis of welding process. So, sources of these equations have to be added.
All parameters in Eqs. (15) must be defined in this paper.
Remove "Eq. " from the numbers of equations.
The source or description of the methods of determination of thermophysical parameters listed in Table 3 should be added.
Figure 2b is not referenced in the text.
Specify the unit of temperature presented in Figure 2b.
Specify (a) and (b) in caption of the figure 2.
Line 133: Fig. 3b. Figures should be placed in the manuscript in order of appearance in the text. So figure 3 should be placed in the text before figure 2. Furthermore, it is suggested to place the figures in the section with first quotation.
lines 152, 154: What does t8/5 field mean?
Units should be specified in figure 3.
Add a number to the equation in line 173.
Figure 6: It is suggested to rearrange the figure into two rows with enlarged micrographs.
The style of references in section References does not meet the Instructions for Authors.
References are not very numerous in the Introduction section. Adding the latest items in the field of laser welding, especially from MDPI journals, will increase the recognition and impact of the article.
